# Transcriptome Analysis of Hypothalamic-Pituitary-Ovarian Axis Reveals circRNAs Related to Egg Production of Bian Chicken

**DOI:** 10.3390/ani14152253

**Published:** 2024-08-02

**Authors:** Peifeng Li, Qi Zhang, Chengzhu Chu, Binlin Ren, Pengfei Wu, Genxi Zhang

**Affiliations:** 1College of Animal Science, Shanxi Agricultural University, Taiyuan 030032, China; 888lipeifeng888@163.com (P.L.); zhang_qi9606@163.com (Q.Z.); 17554054704@163.com (C.C.); 15535392517@163.com (B.R.); 2Tianjin Key Laboratory of Animal Molecular Breeding and Biotechnology, Tianjin Engineering Research Center of Animal Healthy Farming, Institute of Animal Science and Veterinary, Tianjin Academy of Agricultural Sciences, Tianjin 300381, China; 3College of Animal Science and Technology, Yangzhou University, Yangzhou 225009, China; gxzhang@yzu.edu.cn

**Keywords:** Bian chicken, circRNA, reproduction, hypothalamic–pituitary–ovarian axis

## Abstract

**Simple Summary:**

In our study, we aimed to understand the role of non-coding RNA in egg production by analyzing the transcriptome of chickens. We collected tissues from both low-yielding and high-yielding Bian chickens, focusing on the hypothalamus, pituitary, and ovaries, which are key to reproduction. Using advanced sequencing technology, we identified circular RNAs that differed between high and low egg producers. Our results revealed that these circRNAs are involved in crucial processes like cell development, nervous system function, and protein regulation. We also discovered networks of RNA interactions that may influence egg production. These findings enhance our understanding of the genetic factors that control egg production, potentially leading to improved breeding strategies for better poultry production and health, benefiting farmers and consumers alike.

**Abstract:**

The hypothalamic–pituitary–ovarian (HPO) axis plays a pivotal role in the regulation of egg production in chickens. In addition to the traditional understanding of the HPO axis, emerging research highlights the significant role of circRNAs in modulating the functions of this axis. In the study, we collected hypothalamus, pituitary, and ovarian tissues from low-yielding and high-yielding Bian chickens for transcriptome sequencing. We identified 339, 339, and 287 differentially expressed (DE) circRNAs with *p*_value < 0.05 and |log2 (fold change)| ≥ 1 in hypothalamus, pituitary, and ovarian tissues. The Gene Ontology (GO) enrichment analysis for the source genes of DE circRNAs has yielded multiple biological process (BP) entries related to cell development, the nervous system, and proteins, including cellular component morphogenesis, cell morphogenesis, nervous system development, neurogenesis, protein modification process, and protein metabolic process. In the top 30 Kyoto Encyclopedia of Genes and Genomes (KEGG) pathways, we observed the enrichment of the GnRH signaling pathway in both the hypothalamus and the pituitary, solely identified the GnRH secretion pathway in the pituitary, and discovered the pathway of oocyte meiosis in the ovary. Furthermore, given that circRNA primarily functions through the ceRNA mechanism, we constructed ceRNA regulatory networks with DE circRNAs originating from the GnRH signaling pathway, GnRH secretion, ovarian steroidogenesis, steroid hormone biosynthesis, and the estrogen signaling pathway. Finally, several important ceRNA regulatory networks related to reproduction were discovered, such as novel_circ_003662-gga-let-7b/miR-148a-3p/miR-146a-5p/miR-146b-5p and novel_circ_003538-gga-miR-7464-3p-SLC19A1. This study will contribute to advancements in understanding the involvement of circRNAs in the HPO axis, potentially leading to innovations in improving egg production and poultry health.

## 1. Introduction

Eggs are a highly nutritious food source, rich in essential nutrients, such as high-quality protein, vitamins (A, and B12), minerals (iron, and zinc), and essential fatty acids [1,2]. They play a vital role in human nutrition by providing a complete and digestible protein source, making them beneficial for overall health. Research has shown that regular consumption of eggs can contribute to improved nutrient intake and may even have positive effects on cardiovascular health and weight management [3]. Globally, egg consumption has tripled over the past 40 years, reflecting the increasing recognition of eggs as an inexpensive source of high-quality protein and essential nutrients [2]. This growth trend is expected to continue, fueled by expanding urbanization, changing consumer preferences for protein-rich diets, and increasing awareness of the nutritional benefits of eggs.

The hypothalamic–pituitary–ovarian (HPO) axis governs the reproductive processes in hens, influencing follicle selection, development, atresia, and ovulation [4]. The production of eggs in hens involves a series of hormonal changes orchestrated by the HPO axis and gonadotropin-releasing hormone (GnRH) is a central regulator of the HPO axis [5]. Follicle-stimulating hormone (FSH) and luteinizing hormone (LH), which are stimulated by GnRH from the pituitary gland, play crucial roles in regulating the endocrine function and gamete maturation in the gonads, especially pertinent to egg production in hens [6,7]. In addition, gonadotropin-inhibitory hormone (GnIH) [8], Estradiol (E2) [9], and Prolactin (PRL) [10] of the HPO axis were also closely associated with egg production in hens.

Recently, non-coding RNAs were found to contribute to the intricate genetic regulatory networks of genes associated with broodiness, egg production, and ovarian follicle development in poultry [4]. Circular RNAs (circRNAs), as non-coding RNAs, are biologically functional nucleic acid molecules generated through the process of back-splicing from precursor mRNA in living organisms [11,12]. These have also attracted the attention of researchers. Shen et al. [13] used RNA sequencing to profile circRNAs in theca cells from three types of follicles, specifically small yellow follicles (SYF), the smallest hierarchical follicles (F6), and the largest hierarchical follicles (F1), and they identified a total of 14,502 circRNAs in all theca cells, with 5622 widely distributed in all stages of development. Another study found that circRNA_0320 and circRNA_0185 might be involved in granulosa cell differentiation and follicle development of chickens by interacting with miR-143-3p, a miRNA targeting the follicle-stimulating hormone receptor (FSHR) [14]. Wang et al. [15] performed the RNA-seq of ovarian tissues from Lohmann hens and Chengkou Mountain chickens under the laying peak period and found that novel_circ_001023 and novel_circ_005081, derived from MAPK3 and EGFR genes, may be closely related to reproduction. Moreover, novel_circ_000985 and novel_circ_002475 may influence egg production in layers through the ceRNA regulatory network.

Most research has only focused on a single tissue, while systematic studies on circRNA in the HPO axis have not been reported yet. We took Chinese native Bian chickens as the experimental subjects and collected the hypothalamus, pituitary, and ovarian tissues from individuals with high and low egg production for transcriptome sequencing analysis, aiming to systematically screen out reproductive-related circRNAs in the HPO axis.

## 2. Materials and Methods

### 2.1. Animals

The Bian chickens used in the study were selected from the Bian chicken breeding farm of Shanxi Agricultural University. This study recorded the number of eggs laid by Bian chickens at 228 days old, which is a time point when the chickens have reached their peak egg production and maintained this level for one month. Fifteen high-yielding layer hens with uniform body sizes were selected and then randomly divided into three groups of five hens each. Hypothalamus, pituitary, and ovarian tissues were collected and the same samples collected from each group were pooled. Before collecting the ovarian cortex tissue, we removed the larger follicular cells from the ovary. The same method was used for grouping and sampling chickens with low egg production. The statistical results showed that the number of eggs laid by high-yielding hens was 45.33 ± 2.75 g, while that by low-yielding hens was 26.80 ± 3.08 g. The Independent Sample *t*-test revealed a significant difference (*p* < 0.01) in egg production between the two groups.

The samples in each group are named as follows: hypothalamus of high-egg-production chickens (HH), pituitary of high-egg-production chickens (HP), ovarian tissue of high-egg-production chickens (HO), hypothalamus of low-egg-production chickens (LH), pituitary of low-egg-production chickens (LP), and ovarian tissue of low-egg-production chickens (LO).

### 2.2. RNA Extraction, Library Construction, and Sequencing

Total RNA was extracted using Trizol reagent kit (Invitrogen, Carlsbad, CA, USA) according to the manufacturer’s protocol. RRNAs were then removed after RNA quality was assessed on an Agilent 2100 Bioanalyzer (Agilent Technologies, Palo Alto, CA, USA) and checked using RNase-free agarose gel electrophoresis. The enriched RNAs were then fragmented into short fragments by using fragmentation buffer and reverse-transcribed into cDNA with random primers. Second-strand cDNA was synthesized by DNA polymerase I, RNase H, dNTP (dUTP instead of dTTP), and buffer. Next, the cDNA fragments were purified with QiaQuick PCR extraction kit (Qiagen, Venlo, The Netherlands), end-repaired, poly(A) added, and ligated to Illumina sequencing adapters. Then, UNG (Uracil-N-Glycosylase) was used to digest the second-strand cDNA. The digested products were size selected by agarose gel electrophoresis, PCR amplified, and, finally, sequenced using Illumina HiSeqTM 4000 (Illumina, Inc., San Diego, CA, USA) by Gene Denovo Biotechnology Co. (Guangzhou, Guangdong, China).

### 2.3. Bioinformatics Analysis

Raw reads were acquired from the sequencing machines and, subsequently, refined using fastp [16] (version 0.18.0) to produce high-quality clean reads. These clean reads were then mapped to the ribosome RNA (rRNA) database utilizing Bowtie2 [17] (version 2.2.8), and, subsequently, the rRNA-mapped reads were eliminated. The rRNA-removed reads of each sample were then mapped to the reference genome (Ensembl_release105) by HISAT2 [18] (version 2.1.1). After aligning with the reference genome, the reads that could be mapped to the genomes were discarded, and the unmapped reads were then collected for circRNA identification. Next, 20mers from both ends of the unmapped reads were extracted and aligned to the reference genome to find unique anchor positions within the splice site. Anchor reads that aligned in the reversed orientation indicated circRNA splicing, and were then subjected to find_circ [19] to identify circRNAs. A candidate circRNA was called if it was supported by at least two unique back-spliced reads at least in one sample.

Differentially expressed circRNAs (DE circRNAs) were identified using the edgeR package [20] (version 3.12.1) (https://www.r-project.org/) with *p*_value < 0.05 and |log2 (fold change)| ≥ 1. Gene Ontology (GO) is an international standardized gene functional classification system and it includes three ontologies: molecular function (MF), cellular component (CC), and biological process (BP). The source genes of DE circRNAs were mapped to GO terms in the Gene Ontology database (http://www.geneontology.org/). Gene numbers were then calculated for every term and significantly enriched GO terms in source genes compared to the genome background were defined by hypergeometric testing (*p* < 0.05). Kyoto Encyclopedia of Genes and Genomes (KEGG) is the major public pathway-related database [21]. The significantly enriched KEGG pathways were defined with *p* < 0.05. 

We used the software Miranda (version 3.3a) and TargetScan (version 7.0) to predict the targeting relationship of circRNA, miRNA, and mRNA. The correlation (cor) values among circRNA, miRNA, and genes were calculated based on their expression values. Finally, the circRNA-miRNA-mRNA network with cor (miRNA-mRNA) < −0.7 and cor (circRNA-mRNA) > 0.7 remained and was visualized using Cytoscape (version 3.8.2).

## 3. Results

### 3.1. Overview of Sequencing Data

There were 1,639,957,796 raw reads in total, and 1,635,297,404 clean reads accounting for 99.72% were obtained after filtering (Appendix A). The Q20 and Q30 of the clean bases were more than 96.86% (LH-3) and 91.51% (LH-3), respectively. Additionally, the GC contents of clean reads for each sample varied between 44.00% (LP-1) to 46.15% (HO-1). The alignment results of clean reads to the Ribosome RNA (rRNA) showed that the proportion of unmapped reads was higher than 98.93% (HH-1) in each sample (Appendix A). Over 94.37% (LP-3) of the rRNA-depleted clean reads were successfully mapped to the reference genome and the proportion of mapped reads was 4.33–5.63% (Appendix A). The total mapped reads were predominantly located in exons, followed by introns, and, lastly, intergenic regions (Appendix A). The total mapped reads compared to the reference genome of Anchor Reads accounted for 65.07–71.63% and they were, finally, used for circRNA identification (Appendix A).

### 3.2. Differential Expression Analysis

A total of 339, 339 and 287 DE circRNAs were found with *p*_value < 0.05 and |log2 (fold change)| ≥ 1 in the comparisons of LH vs. HH, LP vs. HP, and LO vs. HO (Figure 1A). For the hypothalamic tissue, 171 up-regulated and 168 down-regulated DE circRNAs were found in the group of high-yielding layer hens. Compared with group LP, 155 DE cricRNAs were up-regulated and 184 DE cricRNAs were down-regulated in the group HP. In the ovarian tissue of high-yielding layer hens, we identified 153 up-regulated and 134 down-regulated DE circRNAs. We also found two co-differentially expressed circRNAs (novel_circ_007938 and novel_circ_008624) in the three comparison groups (Figure 1B). Finally, the heatmaps of DE circRNAs were formed with their expression levels and they illustrate the up-regulation and down-regulation relationships of DE circRNAs across different groups (Figure 1C–E).

### 3.3. Functional Analysis for Source Genes of Differentially Expressed circRNAs

GO enrichment analysis for the source genes of DE circRNAs was performed and the top 20 GO terms are presented in Figure 2. Among the top 20 terms of the three tissues (hypothalamus, pituitary, and ovary), we found that BP entries accounted for the highest proportion. BP describes the overall biological activities and processes that a gene or gene product participates in, and it is often considered to be of particular importance. The hypothalamus, pituitary, and ovaries are crucial tissues composed of multiple cells and we enriched many BP entries related to cell growth and development in the top 20 GO terms, including cellular component morphogenesis (GO:0032989), cell morphogenesis (GO:0000902), cell morphogenesis involved in differentiation (GO:0000904), cell part morphogenesis (GO:0032990), negative regulation of the cellular process (GO:0048523), and positive regulation of the cellular process (GO:0048522). We also enriched some BP entries related to “nerves” for the source genes of DE circRNAs in hypothalamus and pituitary tissues, for instance, nervous system development (GO:0007399), neurogenesis (GO:0022008), neuronal stem cell division (GO:0036445), neuroblast division (GO:0055057), neuron projection morphogenesis (GO:0048812), and neuron development (GO:0048666). Although the development of the nervous system itself does not directly involve hormone synthesis and secretion, it is an important regulator of this process. Peptide/protein hormones are a subclass of proteins that play crucial roles in regulating biological processes through their ability to act as chemical messengers and some protein-related BP entries were found in pituitary and ovary tissues, such as the cellular protein modification process (GO:0006464), protein modification process (GO:0036211), cellular protein metabolic process (GO:0044267), and protein metabolic process (GO:0019538).

KEGG pathway enrichment analysis was conducted for the source genes of DE circRNAs and the top 30 pathways are shown in Figure 3. Among the top 30 pathways, we enriched the GnRH signaling pathway in both the hypothalamus and pituitary, identified the GnRH secretion pathway solely in the pituitary, and discovered oocyte meiosis in the ovary. These pathways directly regulate ovarian development or ovulation. In addition, we collectively or individually enriched some important pathways closely related to reproduction in the three tissues, such as growth hormone synthesis, secretion, and action; the MAPK signaling pathway; the Notch signaling pathway; the Wnt signaling pathway; the estrogen signaling pathway; and the cAMP signaling pathway.

### 3.4. CeRNA Regulatory Network of HPO Axis

CircRNAs are capable of recruiting miRNAs to modulate the expression of target genes, constituting one of the primary regulatory mechanisms employed by circRNAs. We further selected the differentially expressed circRNAs corresponding to the source genes enriched in the GnRH signaling pathway, GnRH secretion, and the estrogen signaling pathway in the three comparison groups, as well as the pathways progesterone-mediated oocyte maturation and ovarian steroidogenesis in LO vs. HO. We obtained a total of 14 source genes, corresponding to 19 differentially expressed circRNAs. The circRNA–miRNA–mRNA networks were predicted for them and then filtered according to cor (miRNA-mRNA) < −0.7 and cor (circRNA-mRNA) > 0.7. Finally, 457 circRNA–miRNA–mRNA relationship pairs were obtained (Figure 4), including five DE circRNAs: novel_circ_003662, novel_circ_006522, novel_circ_006398, novel_circ_009397, and novel_circ_003538.

## 4. Discussion 

Eggs are a vital component of the global food supply, offering a rich source of high-quality protein and essential nutrients necessary for human health. Their affordability and nutritional value make eggs a crucial food item worldwide, contributing to food security and dietary needs [22]. The production of eggs is significantly influenced by the HPO axis, which regulates reproductive functions through hormonal interactions [23]. This axis ensures the optimal production of eggs by coordinating the release of hormones that control ovulation and egg formation. Recent research has underscored the role of non-coding RNAs, especially circular RNAs (circRNAs), in regulating the HPO axis [11]. CircRNAs have been found to modulate gene expression and impact ovarian function and follicle development, which are critical for efficient egg production [19]. Understanding the involvement of circRNAs in the HPO axis could lead to innovations in improving egg production and poultry health.

In the study, we collected hypothalamus, pituitary, and ovarian tissues from low-yielding and high-yielding Bian chickens for transcriptome sequencing, aiming to identify key circRNAs associated with egg production. Through GO functional enrichment analysis for source genes of DE circRNAs, we identified many BP entries associated with cell development in the hypothalamus, pituitary, and ovarian tissues. Cellular development plays a fundamental role in the proper functioning of endocrine cells, which are responsible for hormone synthesis and secretion. This development is critical for the regulation of various physiological processes, including reproduction [24]. We also enriched some BP entries related to nerve systems. The hypothalamus produces releasing and inhibiting hormones that control the secretion of hormones from the pituitary gland. Neurons in the hypothalamus release neurohormones into the hypophyseal portal system, which then travel to the pituitary gland to stimulate or inhibit the release of specific pituitary hormones. This neuroendocrine signaling is vital for processes such as growth, metabolism, and reproduction [25]. Furthermore, some protein-related GO terms are prominently enriched in both pituitary and ovarian tissues. Protein metabolic processes are fundamental in the regulation and function of hormones secreted by the pituitary and ovaries, and they ensure proper pituitary and ovarian hormone synthesis and activity, which are essential for reproductive health [26,27]. Additionally, post-translational modifications of proteins, such as glycosylation, phosphorylation, and ubiquitination, are all essential for the proper functioning of pituitary hormones like LH and FSH [28]. 

KEGG pathway enrichment analysis found that, among the top 30 pathways, the GnRH signaling pathway was enriched in both the hypothalamus and pituitary, with the GnRH secretion pathway uniquely identified in the pituitary, and the estrogen signaling pathway was enriched in the ovary. These KEGG pathways are all closely and directly related to reproduction. In the top 30 KEGG pathways, we have also uncovered several KEGG pathways that are intimately associated with reproduction, such as growth hormone synthesis, secretion, and action; the cAMP signaling pathway; the MAPK signaling pathway; the Notch signaling pathway; and the Wnt signaling pathway.

The growth hormone synthesis, secretion, and action pathway was found in the top 30 KEGG pathways of the hypothalamus and pituitary tissues. In the intricate endocrine system, growth hormone (GH) is synthesized and secreted by the somatotropic cells of the anterior pituitary gland [29]. Its release is tightly regulated by the hypothalamus through two primary hormones: growth-hormone-releasing hormone (GHRH) and somatostatin (also known as growth-hormone-inhibiting hormone, GHIH). GHRH binds to its receptors on somatotrophs, activating adenylate cyclase and increasing cyclic AMP (cAMP) levels, which, in turn, promotes GH synthesis and secretion. Conversely, somatostatin inhibits this pathway by reducing cAMP levels [30,31]. The cAMP signaling pathway was identified in the top 30 KEGG pathways of pituitary tissues. GH has significant effects on reproductive health through its direct actions and its modulation of insulin-like growth factor 1 (IGF-1) production. GH can enhance the sensitivity of ovarian follicles to gonadotropins (FSH and LH), thus promoting follicle maturation and estrogen production. It also could modulate IGF-1, which works synergistically with gonadotropins to stimulate the proliferation and differentiation of granulosa cells, which are essential for follicle development and steroid hormone production [32]. 

The MAPK signaling pathway was enriched in all three tissues. This pathway can modulate the secretion of reproductive hormones and it is known to be involved in the regulation of cellular processes critical for reproductive function, including cell proliferation, differentiation, and survival in the hypothalamus, pituitary, and ovarian tissues [33]. In the pituitary gland, Notch signaling influences the synthesis and secretion of pituitary hormones, such as LH and FSH [34,35]. In the ovaries, the Notch pathway plays a role in modulating processes like folliculogenesis, steroidogenesis, and ovulation, thereby affecting overall reproductive health and fertility [36]. The Wnt signaling pathway was found in the top 30 of ovarian tissues. Activation of the Wnt pathway in the context of the ovaries influences various aspects of folliculogenesis, oocyte maturation, and hormone production, which are essential for reproductive health [37]. Furthermore, the Wnt/β-catenin pathway has been implicated in ovarian cancer, where aberrant activation of this pathway can contribute to tumorigenesis and cancer progression in the ovary [38].

The HPO axis is critical in regulating egg production in birds. The hypothalamus secretes GnRH, which stimulates the anterior pituitary gland to release FSH and LH [6,7]. The ovaries, in response to these hormones, produce estrogen and progesterone, which are vital for the maintenance of the reproductive cycle and proper egg formation. This hormonal regulation ensures synchronized follicular development, ovulation, and optimal reproductive performance.

Novel_circ_003662 has the most connection points in the ceRNA network. Its target miRNA gga-let-7b was found to be differentially expressed between sexually mature and immature chicken ovaries. Gga-let-7b is involved in the development and differentiation of chicken ovarian cells, suggesting a role in reproductive processes [39]. Gga-miR-148a-3p has been found to be involved in follicle development and selection in chickens [40]. Another two (gga-miR-146a-5p and gga-miR-146b-5p) are associated with the regulation of lipid metabolism in chickens, which is crucial during the egg-laying stage [41]. In addition, a study showed that the transcript, as well as the protein for SLC19A1, were present in buffalo oocytes and all the pre-implantation embryo stages [42]. Therefore, we speculated that the novel_circ_003538-gga-miR-7464-3p-SLC19A1 axis also played an important role in the reproduction of chickens. Finally, the ceRNA networks of novel_circ_006522, novel_circ_006398 and novel_circ_009397 could also provide some reference.

## 5. Conclusions

In this study, we collected hypothalamic, pituitary, and ovarian tissues from low-yielding and high-yielding Bian chickens for transcriptome sequencing. We sequenced tissues from low- and high-yielding Bian chickens, identifying differentially expressed circRNAs. Functional analysis revealed the involvement of DE circRNAs in cell development, the nervous system, and proteins. Pathway analysis found GnRH signaling and oocyte meiosis processes. We also constructed a ceRNA network from key reproductive pathways, discovering important reproductive-related ceRNA networks including novel_circ_003662-gga-let-7b/miR-148a-3p/miR-146a-5p/miR-146b-5p and novel_circ_003538-gga-miR-7464-3p-SLC19A1. This study advances our understanding of circRNAs’ role in the HPO axis, potentially improving egg production and poultry health.

## Figures and Tables

**Figure 1 animals-14-02253-f001:**
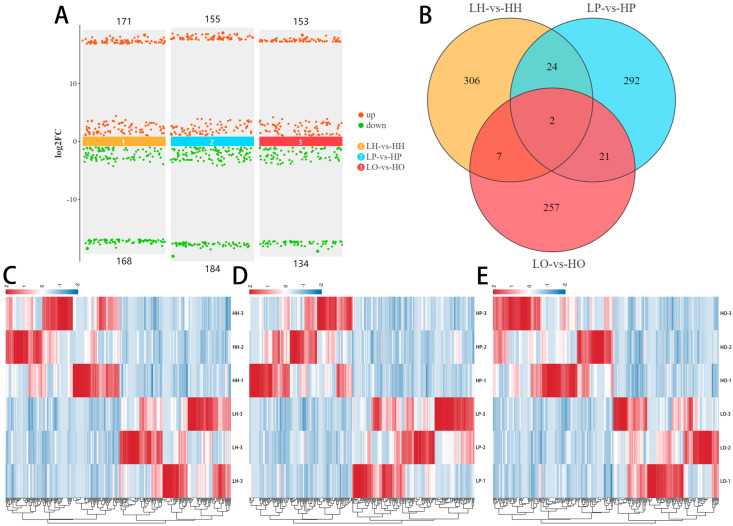
Statistics of DE circRNAs. (**A**) A total of 339, 339 and 287 DE circRNAs were found in hypothalamic, pituitary, and ovarian tissues; (**B**) two co-differentially expressed circRNAs were found in the three tissues; (**C**–**E**) heatmaps showed the up-regulation and down-regulation relationships of DE cricRNAs.

**Figure 2 animals-14-02253-f002:**
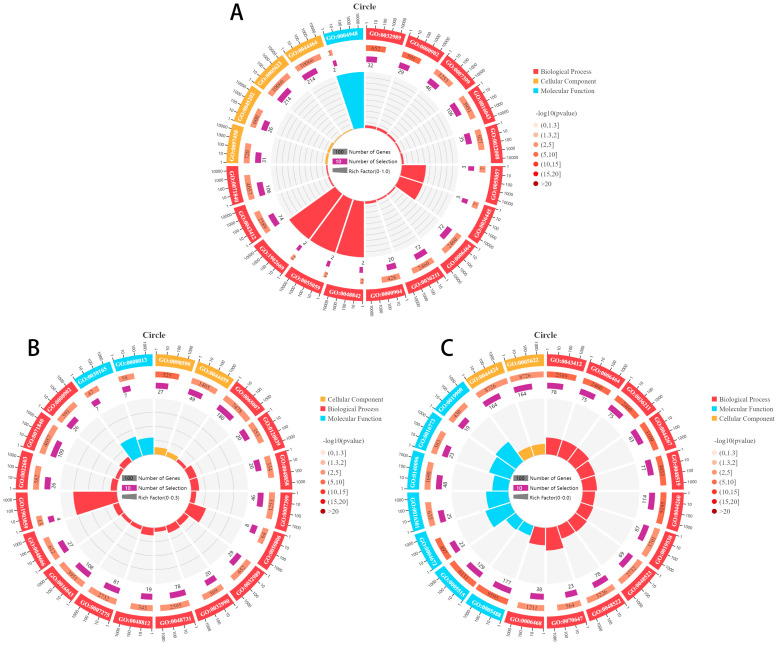
BP terms accounted for the highest proportion in the top 20 GO terms. (**A**) Top 20 GO terms for source genes of DE cricRNA of LH vs. HH; (**B**) top 20 GO terms for source genes of DE cricRNA of LP vs. HP; (**C**) top 20 GO terms for source genes of DE cricRNA of LO vs. HO.

**Figure 3 animals-14-02253-f003:**
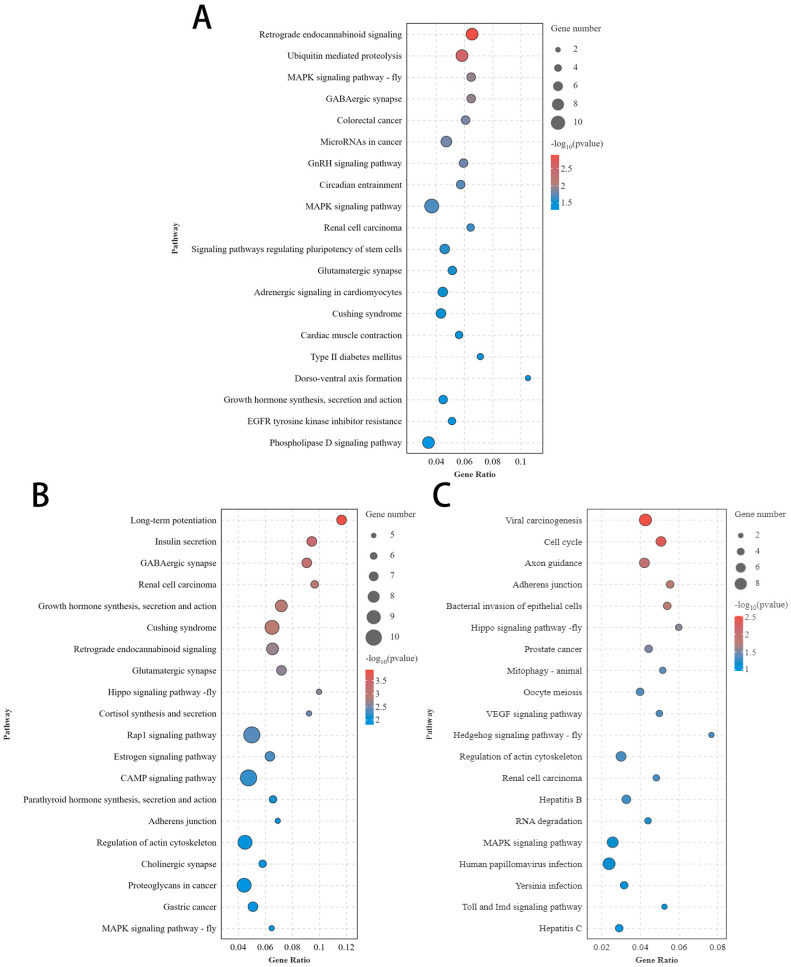
The KEGG pathway enriched some important pathways closely related to reproduction. (**A**) Top 30 KEGG pathways for source genes of DE circRNAs of LH vs. HH; (**B**) top 30 KEGG pathways for source genes of DE circRNAs of LP vs. HP; (**C**) top 30 KEGG pathways for source genes of DE circRNAs of LO vs. HO.

**Figure 4 animals-14-02253-f004:**
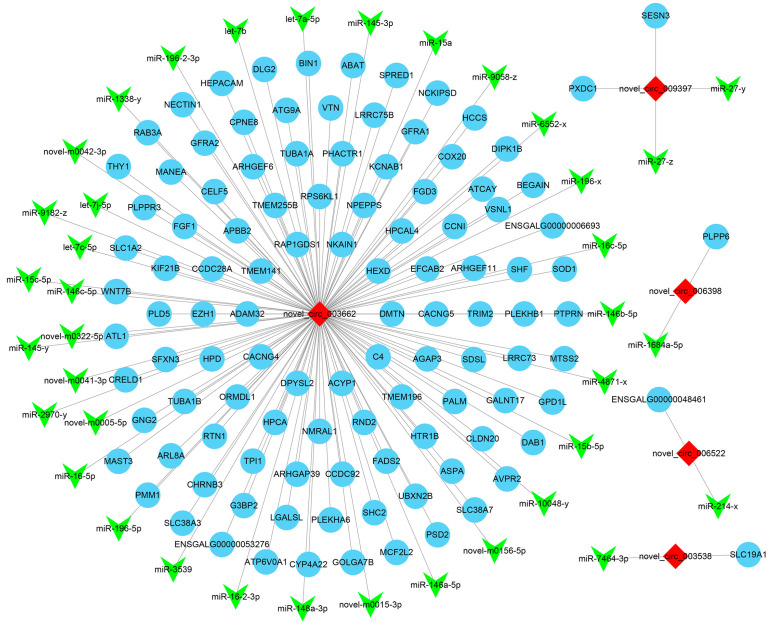
Five important DE circRNAs were identified in the circRNA–miRNA–mRNA interaction network.

## Data Availability

Data are contained within the article and Appendix A.

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
