# Peer review of "Transcriptome Analysis of Hypothalamic-Pituitary-Ovarian Axis Reveals circRNAs Related to Egg Production of Bian Chicken"

_animals, 2024, doi:10.3390/ani14152253_

Round 1
Reviewer 1 Report
Comments and Suggestions for Authors
This study sequenced circRNAs in the hypothalamus, pituitary and ovary of high- and low-laying hens, and the results of the analysis are of some significance for understanding the mechanisms by which circRNAs regulate chicken egg production. However, some details remain to be resolved before publication:
1. Abstract: Whether to use “fold change ≥ 2” or “|Log2(fold change) ≥ 2|” needs further confirmation.
2. Abstract: Full names should be provided for the first occurrence of the terminology, and full names are not required for recurring terminology, so please check the full text.
3. Materials and methods: It is suggested to remove the sentence “The Bian chicken is an eminent native Chinese breed”.
4. Materials and methods: The criteria for classifying high- and low-laying hens should be described in detail. Is it the number of eggs laid from the start of lay to 300 days or at a certain stage? What is the average number of eggs in the high- and low-laying groups? Is there a significant difference in egg production between the high and low laying groups?
5. Materials and methods: How were the ovarian samples collected? Was only the ovarian stroma portion collected or were follicles and ovaries collected together. Further clarification is needed.
6. Results: lines 206-210, mixed up with the discussion, suggest rewording.
7. Results: line 215, how were the 19 DEcircRNAs screened?
8. Results:line 205-220, it is recommended to highlight the regulatory relationships of the hypothalamus, pituitary, and ovaries.
9. Conclusion: Appropriate streamlining is recommended.
Comments on the Quality of English Language
Moderately modify the English language of the entire manuscript, and suggest sending it to native English speakers to polish and edit the language.
Author Response
Dear editor and reviewer:
Thank you very much for your suggestions on the article. We have tried our best to answer each comment point by point and also made corresponding modifications in the article. These changes have been highlighted and would greatly improve the article.
Thank you again for your time and effort!
Sincerely yours,
Pengfei Wu
Reviewer 1:
Comments and Suggestions for Authors
This study sequenced circRNAs in the hypothalamus, pituitary and ovary of high- and low-laying hens, and the results of the analysis are of some significance for understanding the mechanisms by which circRNAs regulate chicken egg production. However, some details remain to be resolved before publication:
- Abstract: Whether to use “fold change ≥ 2” or “|Log2(fold change) ≥ 2|” needs further confirmation.
Response 1 Thank you. We examined the tables of differentially expressed (DE) circRNAs and found that there were DE circRNAs with |log2(fc)| < 2 (as indicated by the arrow in the image below). Therefore, I modified it in the article to | log2 (fc)| ≥ 1.
- Abstract: Full names should be provided for the first occurrence of the terminology, and full names are not required for recurring terminology, so please check the full text.
Response 2 Thank you. I have checked the entire text and have retained both the abbreviation and the full text in the abstract section. Additionally, the abbreviation that first appears in the main body of the paper has retained the full text.
- Materials and methods: It is suggested to remove the sentence “The Bian chicken is an eminent native Chinese breed”.
Response 3 Thank you. I have removed this description and revised this sentence.
- Materials and methods: The criteria for classifying high- and low-laying hens should be described in detail. Is it the number of eggs laid from the start of lay to 300 days or at a certain stage? What is the average number of eggs in the high- and low-laying groups? Is there a significant difference in egg production between the high and low laying groups?
Response 4 Thank you. This study recorded the number of eggs laid by Bian chickens at 228 days old, which is a time point when the chickens have reached their peak egg production and maintained this level for one month. The statistical results showed that the number of eggs laid by high-yielding hens was 45.33±2.75g, while that by low-yielding hens was 26.80±3.08g. The Independent Sample T-test revealed a significant difference (P < 0.01) in egg production between the two groups.
I have added these data and analysis in the article. Line 81-92.
- Materials and methods: How were the ovarian samples collected? Was only the ovarian stroma portion collected or were follicles and ovaries collected together. Further clarification is needed.
Response 5 The tissue we are collecting is the ovarian cortex, and prior to this, we have removed the larger follicular cells from the ovary. And I have added this information in the Materials and Methods section and highlighted it. Line 87-88.
- Results: lines 206-210, mixed up with the discussion, suggest rewording.
Response 6 Thank you. I have changed the sentence. Line 213-214
- Results: line 215, how were the 19 DEcircRNAs screened?
Response 7 We selected reproduction-related pathways such as GnRH signaling pathway, GnRH secretion and Estrogen signaling pathway, and found a total of 14 genes enriched in them. These 14 genes correspond to 19 differentially expressed circRNAs in all the three comparison groups. Line 215-219
- Results: line 205-220, it is recommended to highlight the regulatory relationships of the hypothalamus, pituitary, and ovaries.
Response 8 Thank you. We have provided a detailed description of the regulatory relationship of the hypothalamus, pituitary, and ovaries in the discussion, line 299-305
- Conclusion: Appropriate streamlining is recommended.
Response 9 Thank you. I have made some modifications to the discussion section and highlighted them.
Comments on the Quality of English Language
Moderately modify the English language of the entire manuscript, and suggest sending it to native English speakers to polish and edit the language.
Response 10 Thank you. We have made some revisions to the entire text.
Reviewer 2 Report
Comments and Suggestions for Authors
The manuscript Li and others reported identification of circRNAs that are related with the transcriptome of HPO axis in egg production of Bian chicken. It is potentially important to better understand the molecular circuitry that can contribute to better egg production. The manuscript is, however, not explicit enough to appreciate the value of their findings as is.
1. It is unclear what made the authors focus on the circRNAs? Only one reference was cited – line 58. Need more extensive coverage on circRNAs with sufficient references. How they are generated and how they target ceRNAs, etc.
2. How is circRNAs generation regulated? Need explanation and references. Differential expression of mRNA is sufficient and the circRNA will autonomously be generated? If not, what controls this step?
3. They reported that the novel_circ_003662 is identified with most of the potential target ceRNAs. What is the feature of the sequence that confers such a broad targeting?
4. The authors assumed that circRNA is to sponging miRNA, but there are other mechanisms that circRNAs affect their targets. Need to be covered.
5. Line 161: Hormones are a subclass of proteins that play crucial roles in …
Not all the hormones are proteins.
6. All the figure legends need to be described better. They need to contain summaries of the main observations for each figure.
Comments on the Quality of English LanguageLine 198: “Growth hormone synthesis, secretion and action, MAPK signaling … “
It is confusing. Revise: “synthesis, secretion, and action of Growth hormone, MAPK signaling, …”
Author Response
Dear editor and reviewer:
Thank you very much for your suggestions on the article. We have tried our best to answer each comment point by point and also made corresponding modifications in the article. These changes have been highlighted and would greatly improve the article.
Thank you again for your time and effort!
Sincerely yours,
Pengfei Wu
- It is unclear what made the authors focus on the circRNAs? Only one reference was cited – line 58. Need more extensive coverage on circRNAs with sufficient references. How they are generated and how they target ceRNAs, etc.
Response 1 Thank you. Circular RNAs (circRNAs) have garnered significant attention in recent years due to their unique properties, modes of biogenesis, and potential roles in gene regulation. We have found some circRNA studies related to chicken egg production through literature review. However, most of these studies have focused on ovarian tissues, and few circRNA studies related to chicken HPO have been found. Therefore, we conducted this study. I also added some descriptions of circRNAs in the article. line 58-60
- How is circRNAs generation regulated? Need explanation and references. Differential expression of mRNA is sufficient and the circRNA will autonomously be generated? If not, what controls this step?
Response 2 Thank you. The generation of circular RNAs (circRNAs) is a highly regulated process that involves a combination of genetic, molecular, and structural factors. Unlike the canonical splicing that leads to linear mRNA production, circRNA biogenesis occurs through a process called back-splicing, where a downstream 5' splice site is joined to an upstream 3' splice site, resulting in a circular RNA molecule.
The differential expression of mRNA alone is not sufficient to ensure the autonomous generation of circRNAs. While higher transcriptional activity can increase the availability of pre-mRNA substrates for both linear splicing and back-splicing, the generation of circRNAs is specifically controlled by the complex factors and mechanisms. The interplay between these factors determines whether back-splicing will occur.
However, this study did not focus on the production process of circRNAs, and only aimed to discover these DE circRNAs. We added a brief description of the characteristics of circRNA in lines 58-60
- They reported that the novel_circ_003662 is identified with most of the potential target ceRNAs. What is the feature of the sequence that confers such a broad targeting?
Response 3 Thank you. We used the softwares, Miranda (version 3.3a) and TargetScan (version 7.0), to predict the targeting relationship of circRNA, miRNA and mRNA. Miranda is a tool that predicts targets based on sequence complementarity and thermodynamic stability. TargetScan predicts miRNA targets based on the evolutionary conservation of miRNA binding sites. We have predicted many target genes for each circRNA according to its sequence characteristics. The correlation (cor) among circRNA, miRNA and gene were further calculated based on their expression. Line 135-139
- The authors assumed that circRNA is to sponging miRNA, but there are other mechanisms that circRNAs affect their targets. Need to be covered.
Response 4 Thank you. We appreciate the reviewer’s insightful comment regarding the diverse mechanisms through which circRNAs can influence their targets. Our study specifically focused on the miRNA sponging activity of circRNAs due to its well-established and extensively documented role in gene regulation. The miRNA sponging mechanism has been widely recognized as one of the primary functions of circRNAs, providing a clear and robust framework for our analysis.
Future studies should indeed explore these additional mechanisms to further elucidate the complex and multifaceted roles of circRNAs in cellular processes. We have noted this in our discussion section and plan to address these aspects in our subsequent research endeavors.
- Line 161: Hormones are a subclass of proteins that play crucial roles in …
Not all the hormones are proteins.
Response 5 Thank you. I have changed the sentence to “eptide/protein hormones are a subclass of proteins”. Line 188-189
- All the figure legends need to be described better. They need to contain summaries of the main observations for each figure.
Response 6 Thank you. I have revised all the figure legends.
Comments on the Quality of English Language
Line 198: “Growth hormone synthesis, secretion and action, MAPK signaling … “
It is confusing. Revise: “synthesis, secretion, and action of Growth hormone, MAPK signaling, …”
Response 7 Thank you. “Growth hormone synthesis, secretion and action”. It is a KEGG pathway, an academic term that can be searched in the following KEGG databases (https://www.kegg.jp/).
Reviewer 3 Report
Comments and Suggestions for Authors
This study investigated the role of circRNAs in the hypothalamic-pituitary-ovarian (HPO) axis of chickens. Transcriptome sequencing identified differentially expressed circRNAs related to cell development, nervous system, and protein processes. Key pathways like GnRH signaling and Oocyte meiosis were enriched, and ceRNA networks were constructed, revealing novel reproductive regulatory networks. The findings would contribute to enhancing egg production and poultry health. However, there are still some modifications that need to be made.
1. The abbreviation of LH, HH, LP, HP, LO, HO etc. should be specified in the abstract.
2. Line 52-55: The subject of the clause in this sentence is not clear. If FSH and LH are intended to be the subjects, the sentence needs to be revised accordingly.
3. Line 56: Add 'the' before 'HPO axis' to make it 'the HPO axis'.
4. Line 62: change “follicle” to “follicles”
5. Line 69: revised “Chengkou Mountain chicken” to “Chengkou Mountain chickens”
6. Line 96: Delete the word 'And'
7. Line 112: changed “rRNA removed” to “rRNA-removed”
8. Line 133-135: Please confirm again the software used for predicting the targeting relationship among circRNA, miRNA and mRNA.
9. Materials and Methods part: Please provide a table showing the egg production of high-yielding and low-yielding hens among the sampled individuals in this study, and perform a significant analysis on the data.
10. Finally, please modify the references according to the requirements of the journal and complete all the details for each reference, such as the page numbers for references 4 and 11.
Author Response
Dear editor and reviewer:
Thank you very much for your suggestions on the article. We have tried our best to answer each comment point by point and also made corresponding modifications in the article. These changes have been highlighted and would greatly improve the article.
Thank you again for your time and effort!
Sincerely yours,
Pengfei Wu
- The abbreviation of LH, HH, LP, HP, LO, HO etc. should be specified in the abstract.
Response 1 Thank you. I have modified some sentences and specified the abbreviations in the abstract.
- Line 52-55:The subject of the clause in this sentence is not clear. If FSH and LH are intended to be the subjects, the sentence needs to be revised accordingly.
Response 2 Thank you. I have revised this sentence as below. Line 50-52
Follicle-stimulating hormone (FSH) and luteinizing hormone (LH), which are stimulated by GnRH from the pituitary gland, play crucial roles in regulating the endocrine function and gamete maturation in the gonads, especially pertinent to egg production in hens
- Line 56:Add 'the' before 'HPO axis' to make it 'the HPO axis'.
Response 3 Thank you. I have added the word. Line 54
- Line 62:change “follicle” to “follicles”
Response 4 Thank you. I have changed the word. Line 62
- Line 69:revised “Chengkou Mountain chicken” to “Chengkou Mountain chickens”
Response 5 Thank you. I have revised the word. Line 69
- Line 96:Delete the word 'And'
Response 6 Thank you. I have deleted the word. Line 99
- Line 112:changed “rRNA removed” to “rRNA-removed”
Response 7 Thank you. I have revised it. Line 114
- Line 133-135:Please confirm again the software used for predicting the targeting relationship among circRNA, miRNA and mRNA.
Response 8 Thank you. We used the softwares, Miranda (version 3.3a) and TargetScan (version 7.0), to predict the targeting relationship of circRNA, miRNA and mRNA. Line 135-136
- Materials and Methods part: Please provide a table showing the egg production of high-yielding and low-yielding hens among the sampled individuals in this study, and perform a significant analysis on the data.
Response 9 Thank you. This study recorded the number of eggs laid by Bian chickens at 228 days old, which is a time point when the chickens have reached their peak egg production and maintained this level for one month. The statistical results showed that the number of eggs laid by high-yielding hens was 45.33±2.75g, while that by low-yielding hens was 26.80±3.08g. The Independent Sample T-test revealed a significant difference (P < 0.01) in egg production between the two groups.
I have added these data and analysis in the article. Line 81-92.
- Finally, please modify the references according to the requirements of the journal and complete all the details for each reference, such as the page numbers for references 4 and 11.
Response 10 Thank you. I have checked all the references and made some modifications.